# Inferring country-specific import risk of diseases from the world air transportation network

Pascal P. Klamser[1,2], Adrian Zachariae[1,2], Benjamin F. Maier[1,2,3,4], Olga Baranov[5,6], Clara Jongen[1,2], Frank Schlosser[1,2], Dirk Brockmann[1,2,7] *

1 Department of Biology, Institute for Theoretical Biology, Humboldt-Universität zu Berlin, Berlin, Germany, 2 Robert Koch Institute, Berlin, Germany, 3 DTU Compute, Technical University of Denmark, Kongens Lyngby, Denmark, 4 Copenhagen Center for Social Data Science, University of Copenhagen, Copenhagen, Denmark, 5 Division of Infectious Diseases and Tropical Medicine, University Hospital, LMU Munich, Munich, Germany, 6 German Center for Infection Research (DZIF), Partner Site Munich, Munich, Germany, 7 Center Synergy of Systems (SynoSys), Center for Interdisciplinary Digital Sciences, Technische Universität Dresden, Dresden, Germany

* dirk.brockmann@tu-dresden.de

**Data Availability Statement:** The software "ImportRisk-v1.0.0" to compute the import risk is available under the Zenodo repository https://doi.org/10.5281/zenodo.7852476.

## Abstract

Disease propagation between countries strongly depends on their effective distance, a measure derived from the world air transportation network (WAN). It reduces the complex spreading patterns of a pandemic to a wave-like propagation from the outbreak country, establishing a linear relationship to the arrival time of the unmitigated spread of a disease. However, in the early stages of an outbreak, what concerns decision-makers in countries is understanding the relative risk of active cases arriving in their country—essentially, the likelihood that an active case boarding an airplane at the outbreak location will reach them. While there are data-fitted models available to estimate these risks, accurate mechanistic, parameter-free models are still lacking. Therefore, we introduce the 'import risk' model in this study, which defines import probabilities using the effective-distance framework. The model assumes that airline passengers are distributed along the shortest path tree that starts at the outbreak's origin. In combination with a random walk, we account for all possible paths, thus inferring predominant connecting flights. Our model outperforms other mobility models, such as the radiation and gravity model with varying distance types, and it improves further if additional geographic information is included. The import risk model's precision increases for countries with stronger connections within the WAN, and it reveals a geographic distance dependence that implies a pull- rather than a push-dynamic in the distribution process.

## Author summary

For the spread of a contagious disease, human mobility puts distant places in proximity and geographically closer targets may be effectively much further away. The worldwide flight network is crucial for long distance travels and the previously proposed 'effective

**Funding:** B.F.M received funding through Grant CF20-0044, HOPE: How Democracies Cope with Covid-19, from the Carlsberg Foundation and was supported as an Add-On Fellow for Interdisciplinary Life Science by the Joachim Herz Stiftung. P.P.K, A.Z, F.S received funding through Grant D81870, COVID-19 Lockdown-Monitor, from Germany's Federal Ministry of Health. The funders had no role in study design, data collection and analysis, decision to publish, or preparation of the manuscript.

**Competing interests:** The authors declare no competing interests.

distance' translates this mobility into a distance measure that correlates with the disease arrival time. We use the effective distance to generate a bottom-up and thus parameter-free distribution process of passengers on the flight network, which takes into account all possible flight routes. This allows us to determine the import probability of a disease. Our 'import risk' model outperforms or matches established mobility models, some of which require calibration with scarce or costly data. In contrast, our approach relies on minimal flight network data, that is the number of planes between airports and their passenger capacities, but not on passenger data. Its bottom-up approach enables future studies on country-specific measures for controlling and containing infected passengers, a challenge with existing models. Thus, the 'import risk' model's strength lies in its data simplicity, this relevance to pandemics, and parameter-free design.

## Introduction

The recent decades have seen a considerable increase in mobility: The worldwide number of passenger cars in use increased by an average of about 4% each year between 2006 and 2015, reaching approximately 1 billion in 2015 [1]. This growth is comparable to the yearly increase in the number of sea containers shipped [2], and the global scheduled air passenger count also experienced an annual growth of about 6% between 2004 and 2019 [3] In essence, the world is becoming increasingly interconnected in terms of passenger mobility, both on a small scale (cars) and a large scale (air traffic), as well as in the import and export of goods. This heightened connectivity facilitates the distribution of goods and people, as demonstrated by the distribution of over 400 invasive species through agricultural imports, which is best predicted by the global trade network [4]. A prime example of unwanted side effects of well-connected regions is the potential for pandemics, accompanied by death, economic damage and the potential stigmatization of survivors, migrants and minorities [5–7]. Already the first plague pandemic that started AD 541 in the Nile Delta of Egypt spread in 8 years across the territories (Mediterranean, Northern Europe and Near East) of 2 affected empires because of the intense commerce in the Roman Empire [6]. Nowadays, the intensified exchange reduces the time until a pandemic reaches all parts of the world to months as for the 2009 H1N1 virus that spread from Mexico in 5 months to all continents [8, 9] or the recent COVID-19 pandemic whose variants spread within a few months across the globe [10–13].

The connection strength between world regions is only partly explained by their geographic proximity. Instead, due to historic geopolitical relations [14, 15] pandemics spread rather along an effective distance that is derived from the world air transportation network (WAN) [16–19], or, if applied on a smaller scale, also from other means of transportation [16, 20]. According to the effective distance, region B is closest to region A if the passenger flow from A to B is greater than to other destinations. An intriguing extension is the multipath effective distance, which enhances the prediction of disease arrival times by considering all paths taken by a random walker on the WAN [17]. The effective distance is regularly used to analyze the impact of mobility on the spread of diseases, as for example for MERS [21], Ebola [22], Zika [23] and most recently COVID-19 [20, 24–26]. While it enables a qualitative estimation of disease arrival times, its applicability is severely restricted when it comes to describing the importation of infected passengers from a specific source to a target. However, these import events are highly relevant for political decision-makers and to enable modeling predictions.

In this work, we describe these import events via the "import probability" $p(B|A)$, which is equivalent to the origin-destination (OD) matrix whose element $T_{BA}$ represents the number of

trips from A to B, with the difference that the probability is normalized by all trips starting in A, i.e. $p(B|A) = T_{BA}/T_A$. There are mobility models that fit the OD matrix, requiring a reference OD matrix as seen in the gravity model [27–31]. Additionally, some models integrate OD matrix-fitted models on a smaller scale with the OD matrix of the global air transportation network, creating a multiscale mobility network to represent all modes of transportation [32, 33]. Note that the multiscale mobility model has been successfully employed to analyze past pandemics [34–36]. Yet, it can be extremely difficult to obtain the OD matrix and most often it is estimated by small surveys [37] or alongside a census [38]. Even for the air transportation network derived from a booking system, the OD is only an approximation since passengers increasingly book directly at the airlines (in 2015 30% of all Lufthansa flights were booked directly which increased to 52% in 2018 [39]) and not via the big GDS (global distribution systems) from which most OD-estimates are derived [40, 41]. This means that to exactly compute the air transportation OD matrix, bookings of all GDSs and about 900 airlines must be purchased/estimated and combined. Thus, models that do not rely on an existing reference OD matrix are important and those either assume an underlying decision process without integrating traffic information as the radiation model [42, 43] or they apply a maximum entropy approach to distribute the unknown OD trips along possible routes of a known traffic network [30, 44, 45]. However, none of the above approaches use the effective distance with its qualitative link to disease propagation and none is based on a mechanistic distribution process on a traffic network. To our understanding, a mechanistic process mimics the detailed movement behavior of the passengers on the traffic network, and neither uses only quantities of and between the locations (gravity and radiation model) nor relies on principles of system in thermodynamic equilibrium (maximum entropy model), in other words it is a bottom-up approach. This approach grants us a mechanistic understanding of the observed patterns, enabling us to investigate how modifications impact passenger distribution. For instance, we can analyze how containment interventions along distribution routes reduce the import probability of infected passengers.

In this work, we introduce the import risk model, based on a distribution process following the shortest path tree of the WAN based on effective distance. This process is combined with a random walker that explores all potential paths within the WAN. We are using WAN data from the year 2014 and compare it to the *Global Transnational Mobility Dataset* from 2014 [40], as a ground truth baseline. Additionally, we investigate the discrepancy to the import risk and alternative mobility models as the gravity [27, 31] and radiation model [43] through multiple comparison measures. We find that the import risk model outperforms the alternative models and improves only slightly when it includes not only WAN information but also the geodetic distance between airports. Lastly, we evaluate the quality of import probability estimation for specific countries and assess if and how the geodesic distance is encoded in the import risk estimate.

## Results

### Relating the WAN, OD-probability and the effective distance

In this work, we introduce the import risk, which estimates the probability of a passenger departing from airport A to conclude their journey at any airport worldwide, even those not directly connected to the origin airport. The estimation is based on the traffic flow of airplanes and the respective maximal passenger capacity between airports, a.k.a. the world air transportation network (WAN), provided by the Official Airline Guide (OAG) [46]. This inference-problem is intriguing because it is much easier to monitor the origin and destination of airplanes, than of passengers with possibly multiple connecting flights until their final

destination. In our study, we use the WAN from 2014 (Fig 1A) and compare the derived import probabilities to a reference dataset. The reference import probability is based on the *Global Transnational Mobility Dataset* (GTN) from 2014 [40, 47], which combines an origin-final-destination dataset from a major global distribution system (GDS) with a tourism dataset from the World Tourism Organization (Fig 1B, see Material and methods for more details on the data). Before introducing the import risk model, we contrast the two datasets, introduce the effective distance [16] and quantify its potential as the base metric for our proposed model.

By comparing the world air transportation network (WAN) with the country-specific reference import probability from the GTN (compare Fig 1A and 1B), we see that the airports connected via direct links belong to countries that also have a high import probability. Nevertheless, due to physical constraints and logistical optimization, not all countries with non-zero import probabilities are directly connected to airports in the source country; instead, they are reached via connecting flights. In the context of import probability, estimates based on geodesic distance and the population of the target country are useful but exhibit limitations in certain scenarios. For instance, the import probability for Italy is approximately 1.4 times greater than that for Germany, even though Germany is geographically closer to Canada and

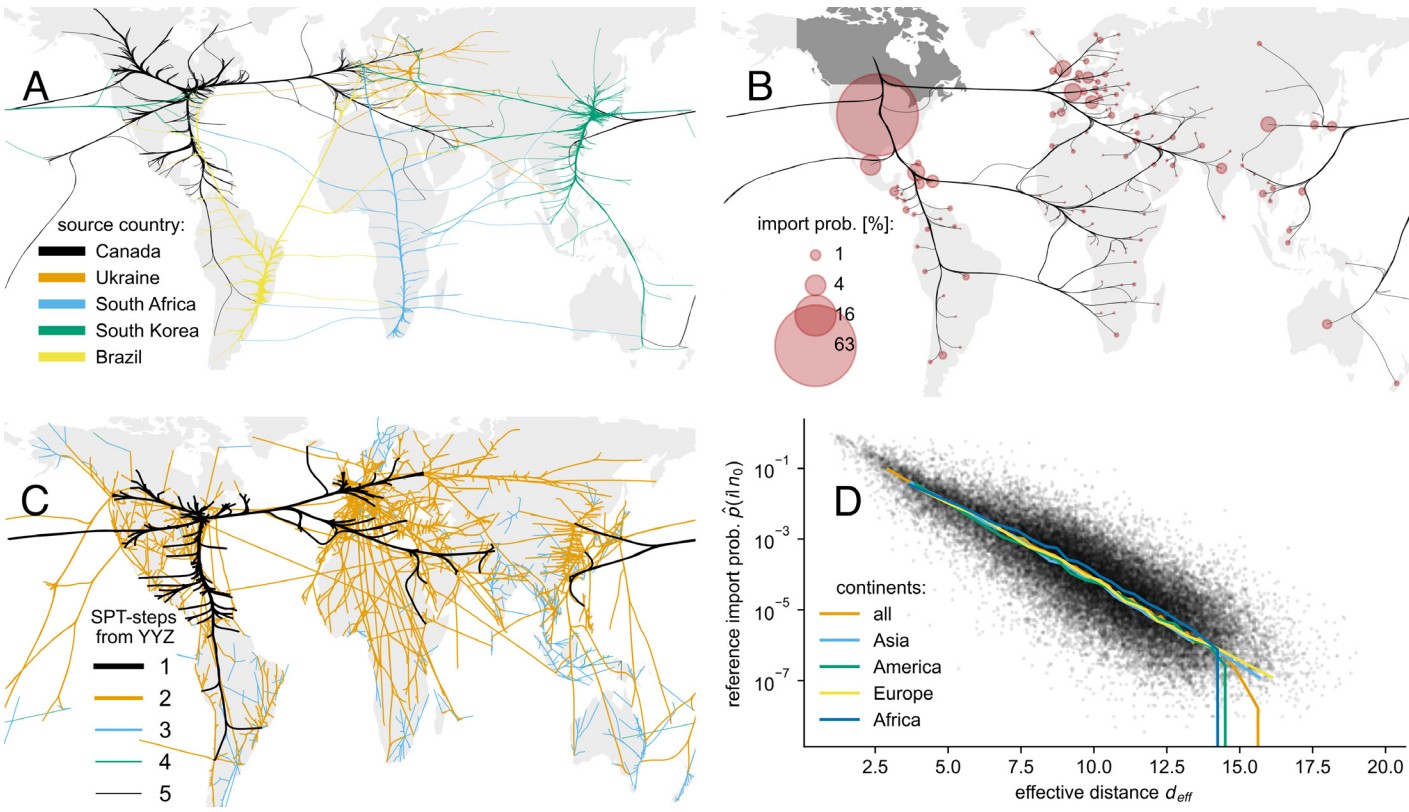

**Fig 1. The relation between WAN, OD-probability, SPT and effective distance. A**: The world air transportation network (WAN) represents the direct flight connections and maximal seat capacities between airports in 2014, here shown for flights starting from five selected countries. It is based on flight-schedule-data. The lines are bundled and do not represent the specific flight route, but illustrate the links to airports abroad. **B**: The reference import probability from Canada to all countries, based on the OD matrix (Origin-Destination) of the *Global Transnational Mobility Data set* [40, 47] in 2014. It combines origin and final-destination trips between countries from the SABRE and the World Tourism Organization (UNWTO). The lines illustrate the connection to the common source country. **C**: Based on the effective distance $d_{\text{eff}} = d_0 - \ln(p)$ a shortest path tree (SPT) is constructed with the largest Canadian airport as source (YYZ: Toronto Pearson International Airport). The link color and thickness shows the hop distance, i.e. number of connecting flights. **D**: exponential decay of the reference import probability (as in **B** but for all countries as source) with the effective distance $d_{\text{eff}}$ (derived from the SPT (**C**) of the WAN (**A**)). Each dot represents a country-country link, the lines are medians including either all source countries or only from a specific continent. Maps are created with geopandas [48].

has a larger population. The effective distance is an alternative network-based distance measure that does not rely solely on direct connections and geographic information [16–19]. Instead, it is based on the passenger flow $F_{ij}$ from $j$ to $i$ and its relationship to the outflow $F_j$ through the transition probability $P_{ij} = F_{ij}/F_j$. Together with a constant distance offset $d_0$, the effective distance between directly connected airports is

$$d_{\text{eff}}(i|j) = d_0 - \ln(P_{ij}) \ . \tag{1}$$

The effective distance between airports without direct connection is the cumulative distance along the shortest path tree (SPT) derived from $d_{\text{eff}}$, as illustrated for the largest Canadian airport (Toronto Pearson Airport, YYZ) in Fig 1C. Note that a distance offset of $d_0 = 0$ would make two routes indistinguishable as long as the product of the transition probabilities along each route is the same, but with $d_0 > 0$ the one route with fewer connecting flights is effectively shorter. Previous studies have demonstrated that the arrival time of diseases in countries exhibits a linear dependence on their effective distance [16–19]. We show that the import probability also correlates with $d_{\text{eff}}$ (Fig 1D), whereby the correlation is higher than for other distance measures (see Fig A in S1 Text). In fact, the import probability decays exponentially with effective distance (linear decay on a semi-log scale in Fig 1D) which can be reproduced in a simplified model for a passenger that travels at a constant effective speed and has a constant exit rate. Therefore, the effective distance seems to be a good representation of the underlying distribution process, and is a promising candidate for the base of our proposed import risk model, to directly estimate the import probability.

## Import risk model

The idea behind the import risk model is a combination of two elements: (i) a random walk with an exit probability of the walker to finish its travel at the current node and (ii) a distribution mechanism derived from the $d_{\text{eff}}$ SPT (Fig 2). The use of a random walk is motivated by Iannelli et al. [17] who could improve the arrival-order prediction of $d_{\text{eff}}$ by including all possible paths. The exit probability enables us to combine the random walk with a distribution mechanism that assigns the likelihood of each node being the final destination, as explained in detail in the second step. In the first step, we use the transition network representation of the WAN and let a random walker start at source $n_0$ and after each step it either exits at the current node $i$ with exit probability $q_i$ or continues to walk. Let us define the walker's *probability to continue walking* to node $n$ given it was at node $n-1$ before and originally started in $n_0$ by

$$S_{n,n-1}(n_0) = P_{n,n-1}(1 - q_{n-1}(n_0)) \ , \tag{2}$$

with $P_{n,n-1}$ as the transition probability from $n-1$ to $n$. Now the probability to walk along a path $\Gamma$ starting at $n_0$ and exiting at $n$ is the probability to continue walking $S_{i,j}$ along each link $(i,j)$ that is part of the path times the exit probability of the final node

$$p(\Gamma) = q_n \prod_{(i,j)\in\Gamma} S_{i,j} \ , \tag{3}$$

where we omitted the explicit dependence on the source $n_0$. Our goal is to describe all possible paths the walker can take from $n_0$ to $n$. We will use the matrix $\mathbf{S}$, whose elements are the probabilities to continue walking $S_{i,j}$. The element $(i,j)$ of the product of the matrix with itself $\mathbf{S} \cdot \mathbf{S} = \mathbf{S}^2$ sums over all paths of length $l = 2$ that end at $i$ and start at $j$. Next, we can define the probability of a walker to exit at $n$ after traversing all paths of length $l$ as

$$p_l(n|n_0) = q_n (\mathbf{S}^l)_{n,n_0} \ . \tag{4}$$

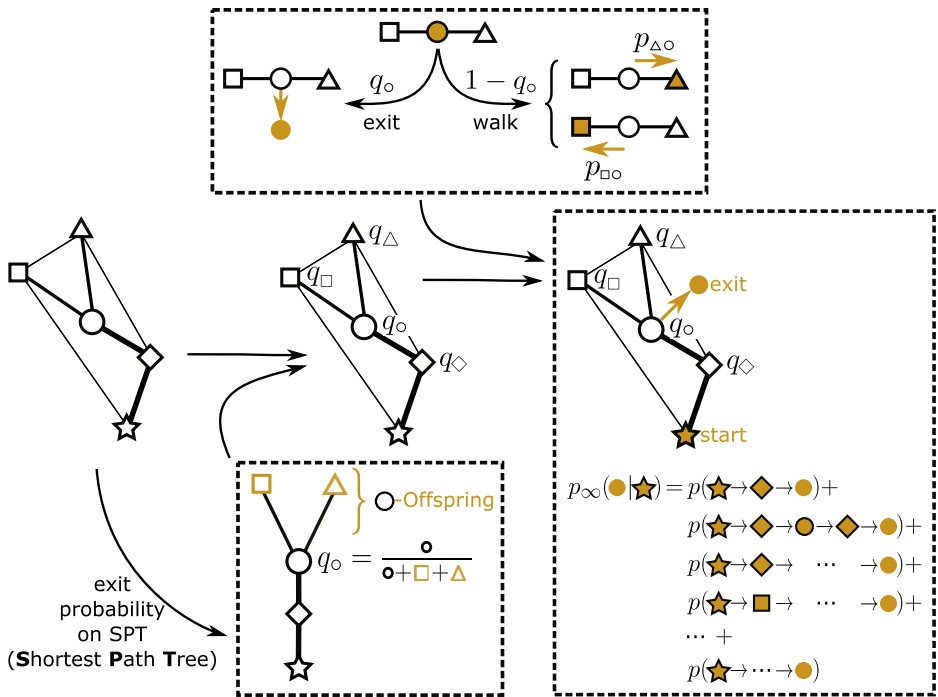

**Fig 2. Import risk scheme.** Starting from the transition network (left) the shortest path tree is computed based on the effective distance (center bottom). Based on the shortest path tree, the exit probabilities $q_\circ = q(\circ|\star)$ are computed. In the formula, the geometric symbols represent the estimated population of the respective node, which can also be distance-weighted (depending on the exact model). A random walk-process with exit probability is defined (top): at each step, the walker either exits the node with prob. $q_\circ = q(\circ|\star)$, or continues walking with prob. $(1 - q_\circ)$. The import risk $p_\infty(\circ|\star)$ (right) is the probability of a walker to exit at node $\circ$ given it started at node $\star$ under consideration of all possible paths.

Finally, the import risk is the probability to exit at $n$ given all paths of all lengths

$$
\begin{aligned}
p_\infty(n|n_0) &= q_n \left( \sum_{l=1}^{\infty} \mathbf{S}^l \right)_{n,n_0} \\
&= q_n((\mathbf{I} - \mathbf{S})^{-1} - \mathbf{I})_{n,n_0} ,
\end{aligned}
\tag{5}
$$

where we used the convergence of the geometric series with identity matrix $\mathbf{I}$.

In the second step, we approximate the exit probability $q_i(n_0)$ that we used above, but did not specify yet. Thereby, we assume that passengers start at source airport $n_0$, travel along the SPT and exit at node $i$ with an exit-probability

$$
q_i(n_0) = \frac{N(i)}{N(i) + N(\Omega(i|n_0))}
\tag{6}
$$

with $N(i)$ as the population at airport $i$ and $\Omega(i|n)$ as the set of all offspring nodes downstream of $i$ on the SPT centered at source $n_0$. Hence, the exit probability at node $i$ is determined by the ratio of the population at node $i$ to the combined populations of all downstream nodes of $i$ on the SPT, inclusive of node $i$.

We estimate the population at airport $i$ using its outflow on the WAN, denoted as $N(i) = F_i$. To aggregate the import probabilities at the country level, we sum the targets and apply a weighted average to the source airports, with population serving as the weighting factor.

To elucidate how additional information about the geographic distance between nodes influences $p_\infty$, we explore two variations of the import risk model: In the variation with "geodesic distance weighted" exit probability the populations in Eq 6 are substituted with $\hat{N}(i|n_0) = N(i)/d_{i,n_0}$, where $d_{i,n_0}$ is the geodesic distance between $i$ and $n_0$. To control for increasing model complexity, we study the "effective distance weighted" exit probability, where $\hat{N}(i|n_0) = N(i)/d_{\text{eff}}(i|n_0)$, i.e. no geographic information is used, but the model structure is equivalent.

**Alternative models.** Numerous alternative models estimate the OD-matrix, from which the import probability can be derived [30, 31, 42, 43, 49–52]. Among those, the gravity [27] and the intervening opportunity [42, 43] model are most widely used. A recent variant of the latter is the radiation model [43]. Although past studies have found that the gravity model outperforms the radiation model at small scale [38, 53, 54], especially the radiation model's good performance at the large scale [38, 54] makes it an interesting model for mobility on the WAN. It was originally conceptualized for commuter flows [43] where the surrounding populations serve as a proxy for possible job opportunities. By estimating an airport's population based on its outflow, we adjust the concept from job opportunities to tourism opportunities. Its derivation from a mechanistic decision process makes it parameter free, and therefore similar and a good comparison to our model. However, it only requires information on the population density and does not integrate flight data.

We compare our model to the gravity model with an exponential and power-law distance dependence and the radiation model (see Material and methods for definitions). These models solely rely on the outflow data from the WAN to estimate the node's population and the geographic locations. To incorporate structural information of the WAN [55], the alternative models are also implemented with the geodesic path distance (the geodesic distance along the SPT) and the effective distance, i.e. there are in total nine alternative models: the radiation model, the gravity model with exponential and with power-law distance decaying function, and each implemented with geodesic, geodesic path and effective distance. The exponents of the six gravity models are fitted to the reference import probability by assigning the best fitting exponent to each of the six comparison measures (Pearson correlation, root-mean-square error, common part of commuters, Kendalls rank correlation and the correlation and RMSE of the logarithmic measures, all defined in Material and methods) and taking their mean value (see Figs B and C in S1 Text). As comparison measures, we have chosen three measures that are related to the absolute error and three that are related to the relative error between estimate and reference.

**Symmetry by returning visitors.** Each of the twelve models provides an estimate for the import probability $p(i|n_0)$, which is used to compute the OD-matrix **T** through multiplication with the corresponding source population $N(n_0)$. By comparing the symmetry of **T** with the reference OD-matrix $\hat{\mathbf{T}}$, we find a much higher and qualitatively different symmetry in the reference data (see Supplementary Note B, Fig D in S1 Text). The high symmetry is likely due to visitors (family, business, tourism, etc.) that dominate the international travel. They return to their home-location after a limited period [56] and only the minority of the travelers are migrants, i.e. stay permanently at the destination. Interestingly, the import risk model has the highest symmetry, but is still less symmetric than the reference data by a factor of 4. Therefore, before conducting a detailed comparison of the estimates, we rectify the import probability estimates by symmetrizing their OD-matrix (by extracting the symmetric part and recalculating the import probability; for further details, refer to Material and methods and Supplementary Note B in S1 Text). This correction can be seen as an alternative version of a doubly constrained model where normally the constraints on in- and out-flow are ensured by an *iterative proportionate fitting* [31].

## Model comparison

In the subsequent analysis, we evaluate the import probability estimates against the reference data through four approaches: (i) a direct comparison and assessment of their medians to identify potential systematic errors, (ii) the application of six distinct goodness-of-fit metrics to assess the individual model's rank and relative performance, (iii) a classification task identifying countries with the highest import risk, particularly relevant in the context of a pandemic and (iv) a correlation study of the arrival time of 20 diseases and SARS-CoV-2 variants.

**Qualitative comparison.** In Fig 3 the import probability estimate $p(i|n_0)$ of each model is compared to the reference import probability $\hat{p}(i|n_0)$. The gravity models exhibit the closest agreement with the reference data when the effective distance is employed, as indicated by the medians (Fig 3, first and second columns). In contrast, the median values of the radiation and import risk models are relatively stable and less influenced by variations in distance metrics or their associated weighting (third and fourth columns). All models overestimate the lowest median import probability (leftmost orange dot in Fig 3), since the estimated import probability is always nonzero, but a large proportion of the lowest reference import probabilities are zero due to the limited observation period and/or an insufficient number of departing passengers. The overestimation of the median import probability is observed up to $p(i|n_0) \leq 10^{-4}$ for both the gravity and import risk models. However, this overestimation is notably absent in the case of the gravity model with an exponential distance decaying function and the effective

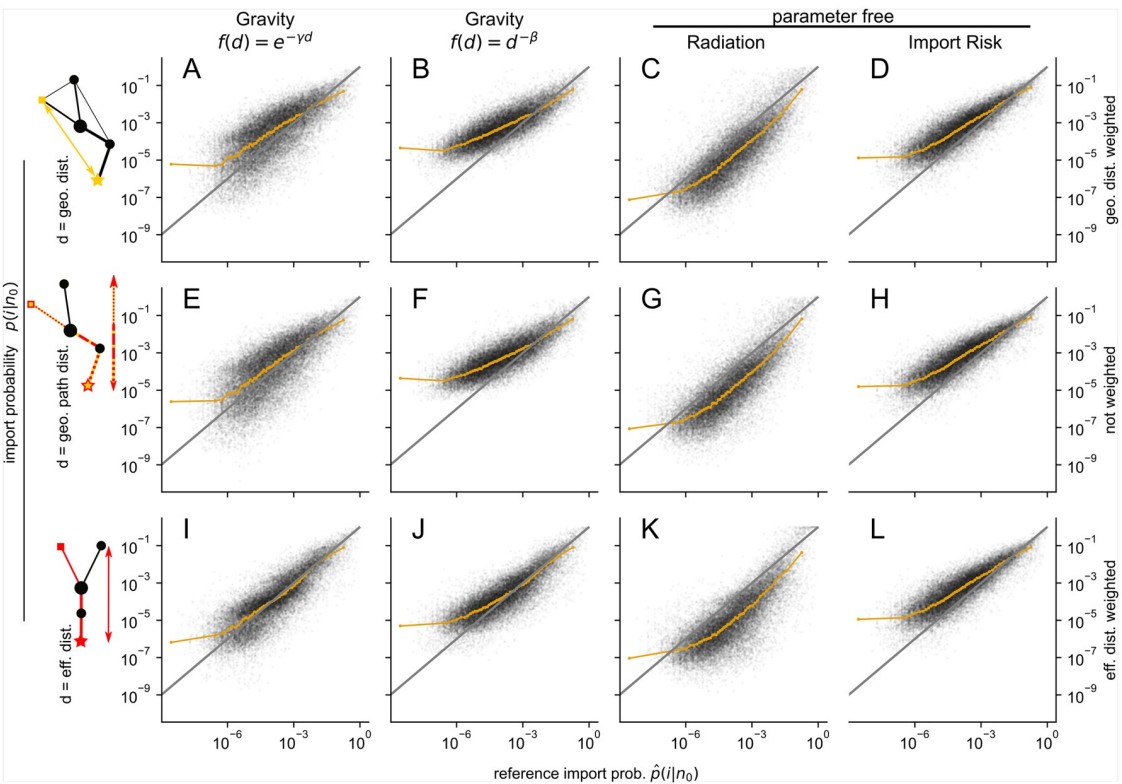

**Fig 3. Estimates of import probability by the gravity model with exponentially (1st column) and power law (2nd column) decaying distance function, the radiation model (3rd. column) and by the import risk model (4th column).** The first three models (1st-3rd column) use as distance the geodesic (1st row), geodesic path (2nd row) and the effective (3rd row) distance. The import risk model is computed from the WAN with the geodesic distance (**D**) or the effective distance (**L**) as a weight for the exit probabilities or without weighting (**H**), i.e. in the last two cases (**H, D**) only WAN information is used. The orange line depicts the median and the gray line is $y = x$ and illustrates perfect mapping.

distance metric (Fig 3I), where the median demonstrates the closest alignment with the reference data. The radiation models (third column) systematically overestimates the highest import probabilities ($p(i|n_0) \gtrsim 10^{-1}$) and consequently underestimates the lower import probabilities.

**Goodness of fit by multiple measures.** We compared each model with the reference import probability via the Pearson correlation, the root-mean-square error (RMSE), and the common part of commuters. These measures are more sensitive to strong links, i.e. large import probabilities, which is important when the emphasis is placed on the countries that are most likely to import passengers. However, if the focus is to get a fair comparison including all links, logarithmic versions of the above measures or rank correlations are more appropriate. Thus, we also quantify the agreement by the correlation and the RMSE of the logarithm of the measures and by Kendall's rank correlation. The three import risk model variations outperform the other models in all but one measure, whereby the variation employing the geodesic

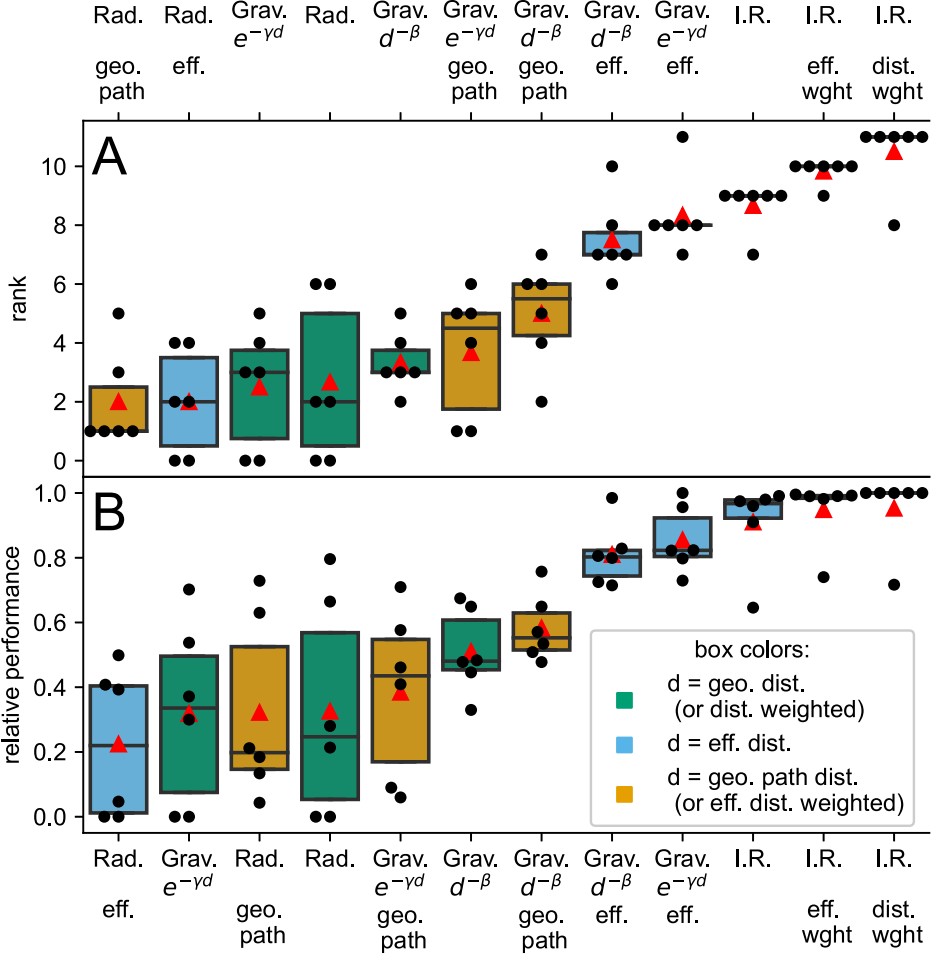

**Fig 4. Rank and relative performance of import risk estimation models.** The different import probability models are compared via their rank (**A**) and relative performance (**B**), with the highest values representing the best approach. The rank and relative performance are shown for each (black dots) of the six comparison measures (corr, logcorr, RMSE, logRMSE, cpc, $\tau_{\text{Kendall}}$) the box illustrates the interquartile range, the horizontal line the median and the red triangle the mean. The colors of the boxes illustrate the different distance measures in use. The outlier measure of the import risk models (I.R.) is the logRMSE, where the gravity models with effective distance are performing best. See Material and methods for definitions of comparison measures and Figs E, F in S1 Text for absolute and detailed relative performance.

distance weighted exit probability performs best (Fig 4A). Following the import risk models, the two gravity models based on effective distance also exhibit strong rankings. In contrast, the remaining models lack consistent high rankings across all six measures and are more evenly distributed within the lower half. This model categorization also holds for the relative performance of the models (Fig 4B), with linear scaling of values in between (see Eq 22). In contrast to the rankings, the median relative performance shows a notable improvement when the gravity models incorporate effective distance. However, among the import risk models, the difference in median relative performance remains marginal.

The only measure where the import risk models are outperformed by the gravity models with effective distance is the logRMSE (Figs E, F in S1 Text). It is expected from the gravity models' good agreement in median import probability with the reference data over wide ranges and the overestimation of low import probability by the import risk model. This overestimation can be reduced by model-modifications that introduce parameters favoring the exit at nodes with large-populations (for details, see Supplementary Note C and Figs G, H in S1 Text). However, we refrain from adding complexity to the model, since its generic nature is its key aspect.

**Classification of ten top risk countries.** In a pandemic context, it is of specific interest to identify the countries with the highest import probability. We analyzed how well the twelve proxy models can classify, if a country is among the ten countries with the highest import probability. Again, the import risk models outperform the other models and the one with geodesic distance-weighted exit probabilities is the top predictor with a sensitivity of 71.1% (Fig 5D). All effective distance-based models have a high sensitivity ($\gtrsim 65\%$), including the radiation model with 66.8% that had the lowest relative performance and second-lowest mean rank (Fig 5I–5K). For these high import probabilities, the import risk models now outperform the

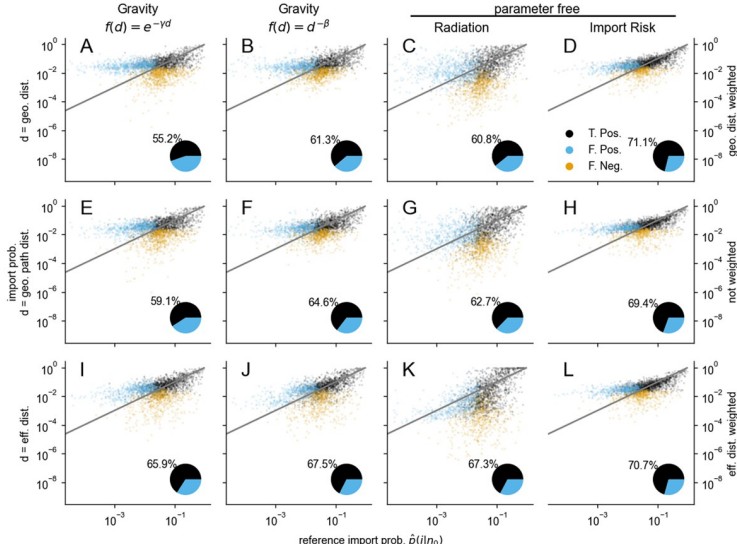

**Fig 5. Classification of the 10 countries with the highest import probability** by the gravity model with exponentially (1st column) and power law decaying (2nd column) distance function, the radiation model (3rd. column) and by the import risk model (4th column). A true or false positive (T. Pos. or F. Pos.) means that the country is or is not among the 10 countries with the highest reference import probability $\hat{p}$. A false negative (F. Neg.) means that it belongs to the reference set but was not detected by the respective model. The pie chart illustrates the sensitivity of the models.

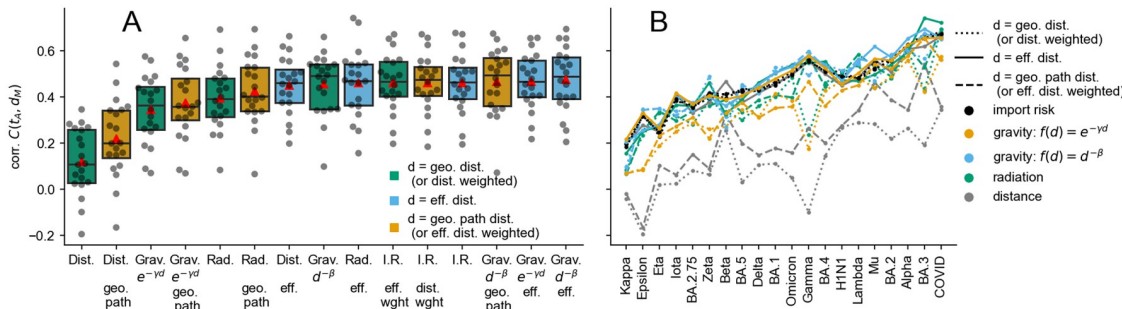

**Fig 6. Correlation analysis: Disease arrival time vs. the effective model distance.** Each model's import probability is converted to an effective distance $d_M(i|n_0) = -\ln(p(i|n_0))$ with $n_0$ as the outbreak country of the respective disease. The correlation results $C(t_A, d_M)$ with the arrival time $t_A(i)$ of the disease in the target country $i$ are grouped by model (**A**) and by the disease (**B**). As comparison distances, the correlation of the geodesic, geodesic path (on the effective shortest path tree) and the effective distance with $t_A$ are shown. Each dot represents a correlation result of the 21 considered outbreaks (H1N1 in 2009, COVID-19 in 2020 and the spread of 18 of its variants in the years 2020–2022).

other models also in terms of RMSE and logRMSE, i.e. the 10 countries at highest risk are not only classified best by the import risk model, but also quantitatively assessed best.

**Disease arrival time.** In our final comparison, we evaluate the correlation between disease arrival times and the estimated import probability from the outbreak country of the disease. Note that the effective distance, which is the base of the import risk model, already has the clear relation to disease arrival times and the import risk model is developed to extend this qualitative relation to a quantitative number of passengers imported, as done in a recent study on the pandemic potential of SARS-CoV-2 variants [11]. However, a qualitative comparison to arrival time is of course possible via the negative logarithm of the import probability for each model, which we refer to as *effective model distance*, which linearly relates [16, 19] to the arrival time $t_A(i|j)$ of a disease

$$d_M(i|j) = -\ln(p_{ij}) \propto t_A(i|j) \qquad (7)$$

with $j$ as the disease outbreak country. The arrival time $t_A(i|j)$ is the number of days between the disease outbreak and the day the first case is reported in the target country $i$. We evaluated the correlation $C(t_A, d_M)$ for the H1N1 pandemic starting 2009 [8], the COVID-19 pandemic starting 2019 [57] and 18 of its variants. Additional to the import probability models, the correlations of the geodesic, geodesic path and effective distance with $t_A$ are included. Our analysis reveals that models employing the effective distance as the distance measure consistently outperform those relying on the geodesic or geodesic path distance (Fig 6A). Interestingly, the gravity model with a power-law decaying distance function consistently performs well, regardless of the specific distance measure employed. We do not observe a specific model that excels exclusively for certain diseases. Instead, we observe similar correlation values for the same disease across models (Fig 6B), which suggests that there is considerable noise on the arrival time $t_A$ that varies between diseases. The noise could be related to the disease specific spreading speed: our assumption, that the outbreak country is the sole source, gets increasingly violated the slower the disease spreads, because other countries become secondary sources. A simple linear regression of the mean correlation $\langle C(t_A, d_M)\rangle$ and the mean arrival time $\langle t_a \rangle$ supports this hypothesis ($r = -0.44$, $p = 0.055$, Fig K in S1 Text).

## Import risk of countries and regions

Having quantified the performance of the import risk model, we now focus on (i) country specific differences in its prediction quality, (ii) possible limitations due to no concept of administrative units (e.g. countries) whose airports are more interconnected and (iii) how the geodesic distance is encoded in the import risk model, i.e. how a distance dependence emerges from WAN information only.

**Country specific performance.** In the import risk approach, we assume minimal knowledge of the system, i.e. only the WAN is known. Consequently, we differentiate countries only via their network properties, one of which is the degree of a node, or more precisely the node strength, since the WAN is a weighted network. It is the simplest metric that is also easily adjustable for the country-level perspective. At the country level, the node strength corresponds directly to the flow out of country $C$

$$F_C = \sum_{n \in C} \sum_{m \notin C} F_{mn} \; . \tag{8}$$

This country-specific characteristic signifies a country's potential to influence the network's structure, since flows from small-outflow countries are diluted by large-outflow countries. From an ecological point of view, the outflow is strongly correlated with the gross domestic product of a country (Fig N in S1 Text). The correlation (logcorr) between the logarithms of the import risk $p_\infty$ and the reference import probability $\hat{p}_\infty$ improves with the outflow of the source country (Fig 7), as illustrated by Great Britain (GB) as the country with the largest outflow in the WAN and Eritrea (ER) as one of the countries with the lowest outflow. The prediction improvement with the country's outflow suggests that the WAN is dominated by large-outflow countries and therefore predictions worsen for countries with lower WAN outflow. However, the prediction improvement is also present in model alternatives that do not use WAN information at all (e.g. gravity with geodesic distance, Fig M in S1 Text). We rule the explanation out that the alternative models show this improvement due to preferential fitting of strong links—and therefore of large-outflow countries—since the models are fitted to the reference data by their import probabilities, which ensures equal weighting among countries. It rather suggests that the mobility behavior in low outflow regions is different, also supported by the sudden performance saturation for countries with a WAN outflow of $F_C \gtrsim 10^6$ (Fig 7 and Fig M in S1 Text). Possibly, their passenger distribution is constrained by additional factors and is limited to the regions in proximity.

There are clear exceptions where the import risk estimation is worse compared to outbreak countries with a similar WAN outflow, as Australia (AU), Israel (IL) and Macao (MO). These countries are connected due to historical relations to specific regions that are either not in their direct neighborhood (European countries for AU and IL) or that are more important than the bare neighborhood would suggest, as Macao that is a special administrative region of China. For Macao the import risk to China is underestimated, which consequently overestimates the import to other countries, and for AU and IL Europe is underestimated which overestimates other regions (Fig 7). AU, IL, and MO serve as examples illustrating that the WAN may not fully encapsulate all relevant information accessible to the import risk model. Another concept that is missing in our methodological approach is the idea of a country or another administrative unit. Instead, it treats airport pairs uniformly, disregarding their country affiliations. Since we know the international flights leaving a specific country from the WAN, we can run a self-consistency analysis, i.e. without the need of reference import probability data.

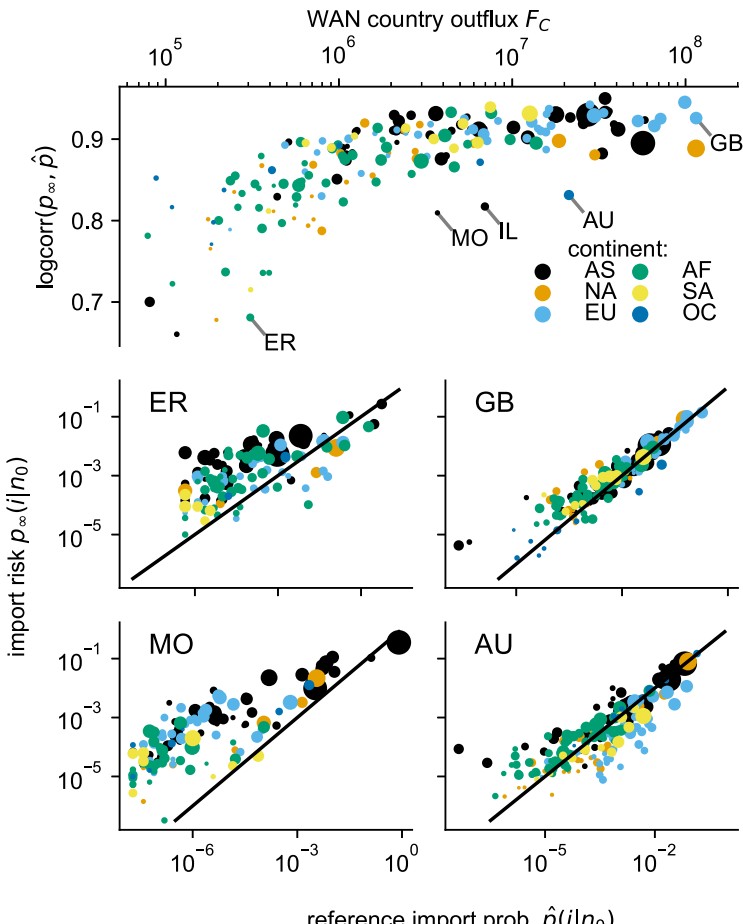

**Fig 7. Source countries' prediction quality and WAN outflow.** The correlation between the logarithm of the import risk and the reference import probability $\text{logcorr} = \text{corr}(\log(p_\infty), \log(\hat{p}))$ improves with the outflow of the respective source country (top). Examples of source countries with particularly low (ER, Eritrea) and high (GB, Great Britain) outflow and log_*corr* are shown with their import risk and reference import risk to target countries (middle row). Countries with exceptionally low log_*corr* measures compared to source countries with a comparable outflow are either historically linked to specific regions as Australia (AU) and Israel (IL) to European countries (lower right panel) or politically as Macao (MO) as a special administrative region of China.

We can estimate the outflow leaving the country $C$ by the import risk model by

$$T_C = \sum_{n \in C} \sum_{m \notin C} p_\infty(m|n) N_n \ .$$

(9)

If we compare it to $F_C$ the WAN flow out of country $C$ (see Eq 8), it turns out that the import risk model systematically overestimates the flow out of a country (Fig I panel A in S1 Text). In fact, the relative error increases with the number of airports belonging to the country (Fig I panel B in S1 Text). Possible explanations for this overestimation include the absence of a country-specific concept within the import risk model and the unintentional inclusion of transit passengers in the population count of airport catchment areas (since we use the outflow as a proxy for the population). However, we can easily correct for this overestimation on country-level analysis, by normalizing the airport population such that the WAN country outflow is recovered.

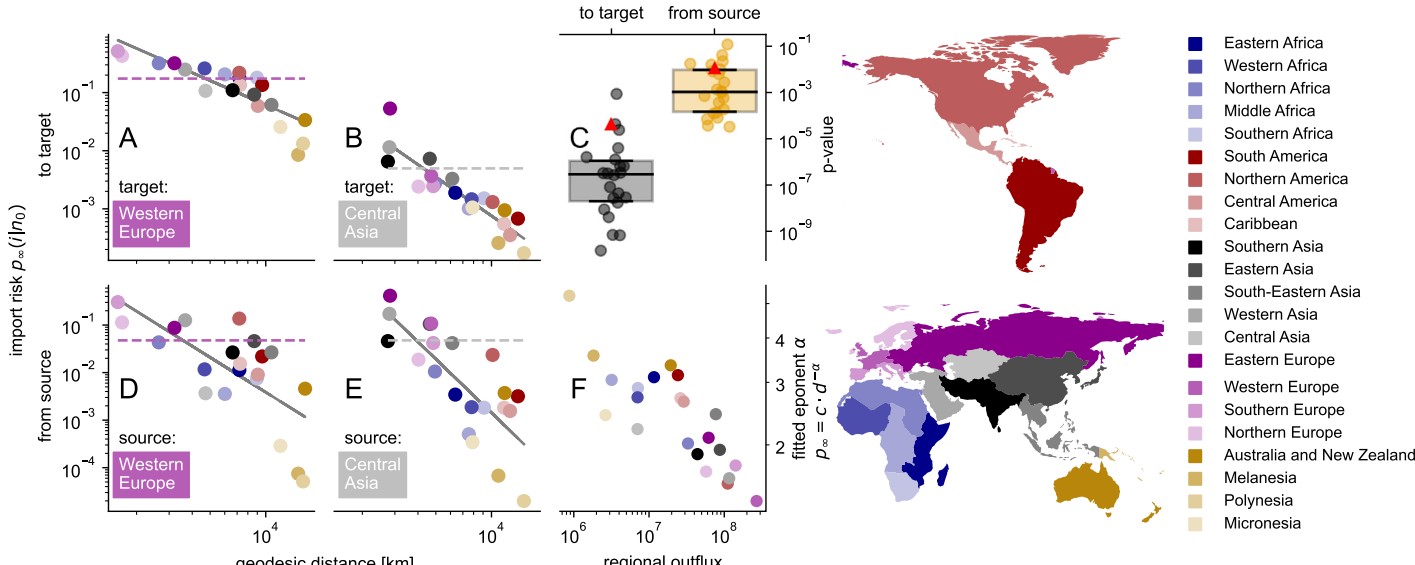

**Fig 8. Import risk aggregated on regional level "to target" vs. "from source" and its geodesic distance dependence.** The geodesic distance between regions predicts the import risk $p_\infty$ to a single target from all sources (**A**, **B**) better than from a single source to all targets (**D**, **E**) as can be seen by the p-values (**C**) of the power law fit $p_\infty(d) = c \cdot d^{-\alpha}$ that is illustrated for each selected examples by a grey line (**A**, **B**, **D**, **E**). The fitted exponent $\alpha$ of the import risk to a single target decreases with the respective regional WAN flow out of the target region (**F**), i.e. the more connected a region, the weaker the import risk decays with distance. The dashed horizontal lines show the average import risk of a single target (**A**, **B**) or a single source (**D**, **E**). The color of the dots corresponds to the depicted world regions (right). Maps are created with geopandas [48].

**Geodesic distance dependence.** The import risk model estimates import probabilities without explicit geodesic-distance information (excluding the variant with distance weighted exit probability). Since classical models have proven distance to be a good predictor for human mobility, we assume that it is encoded in the WAN structure and by consequence in the import risk estimate [58]. To enhance clarity, we aggregate the import risk data across twenty-two world regions. We observe that the import risks to individual targets decrease in a manner resembling a power-law as the geodesic distance to the sources increases (Fig 8A and 8B and Fig L in S1 Text). When we change our perspective and examine the distance-dependence from a single source to all target regions (Fig 8D and 8E), the observed dependence is less consistent with a power-law fit of the form $p_\infty = c \cdot d_{ij}^{-\alpha}$ (Fig 8C). This is surprising, since the import risk is computed via a source-centric view (by computing the exit probability from the shortest path tree originating at each source), which suggests that the distance dependence should be best from one source to its possible targets. A possible explanation is that each target possesses its own attractiveness independent of the source region. This suggests that the distribution dynamics may resemble a pull mechanism rather than a push mechanism. Indeed, we find that the fitted exponent $\alpha$ from the power-law fit decreases as the WAN flow out of the target region increases, which can serve as a proxy for the attractiveness of a region (Fig 8F). In other words, the more attractive a region, the larger the import risks from more distant source regions. The fitted exponent $c$ has a high rank correlation with $\alpha$ ($\tau_{\text{Kendall}} = 0.89$), i.e. also the coefficient is dependent on the attractiveness of the region.

## Discussion and conclusion

Motivated by the import probability's strong dependence on the effective distance, we implemented the import risk model based on the effective distance shortest path tree's exit probability in combination with a random walk on the WAN. As a result, we can infer the passenger trip distribution within the traffic network of their transport vehicle (WAN). When we compare our parameter-free model to variations of established mobility models, we observe that it surpasses the alternatives in most comparison measures. The only exception is where the two parameter-fitted gravity models with effective distance perform the best. The import risk model is the most accurate in determining countries with the highest import probability and is one of the models that correlate best with the time of arrival of 20 diseases, showcasing its importance for epidemic-related problems. However, it systematically overestimates low import probabilities and its performance worsens for countries with a passenger outflow below a million per year. Despite the lack of any explicit geodesic distance information, the import risk model recovers a geodesic distance dependence. This distinction is more prominent when considering all sources to a single target compared to the reverse scenario. We attribute this phenomenon to a target's specific attractiveness, which we estimate using its node strength, i.e. the target's passenger outflow.

The only measure where the gravity models with effective distance outperform the import risk models is the logRMSE. This is likely due to their good agreement over wide ranges of the import probability (Fig 3I and 3J). The import risk model performs poorly with respect to the logRMSE due to its systematic overestimation of low import probabilities. Note, that the second parameter free model, the radiation model, systematically underestimates low import probabilities in the same way as the import risk model does. This is expected, since deviation from the assumptions cannot be corrected by any parameter adjustment. We identified several ways to reduce the import risk's overestimation of low import probabilities by introducing an additional parameter that scales the population of the respective airport, changes the exit probability along the shortest path tree or only the exit probability of specific nodes (for details, see Supplementary Note C and Figs G, H in S1 Text). In conclusion, we find that introducing modifications that enhance the probability of exiting at airports or nodes with large populations mitigates the issue of overestimation. However, we leave this as a possible extension of our model and highlight that it outperformed the other models in all correlation measures, illustrating its high potential.

The radiation model's poor performance can likely be attributed to its initial design, which focused on small-scale commuter flows driven by work opportunities [43], which shows that bottom-up approaches are often limited to their specific use case but can be adapted, such as the extended radiation model [59], which is no longer parameter-free and has similar performance to the gravity model [54]. Interestingly, the radiation model is the only one that does not improve with inclusion of flight network information via the geodesic path or the effective distance (Fig 4). The radiation model's insensitivity to network information can be attributed to the fact that it only extracts rank information from the distance data, resulting in a significant loss of information. The rank representation has the problem that airports that directly follow in their rank with respect to a source airport could be separated by a mountain range or ocean, i.e. the rank difference is minimal but the actual distance immense. This argument holds for any distance information.

We corrected the import probability by the symmetrization of the respective OD-matrices which corresponds to a specific form of a doubly-constrained model. Normally, the constraints only ensure that the out- and inflow of each location corresponds to the observations [31, 52, 54], in contrast, we assume that both equal each other because of returning visitors. We

repeated the model comparison without the correction: it reduced the agreement with the reference data for all but five of the seventy-two model-measure combinations (Fig F in S1 Text), which is in agreement with previous studies that report a better performance of doubly constrained models [54]. Importantly, the import risk model still outperforms the other models if the import probability estimates are not corrected (compare Fig 4 with Fig J in S1 Text). It's crucial to note that the assumption of returning visitors is applicable when visitors and tourists dominate while migrants can be disregarded. However, this assumption may not hold for links between low- and high-income countries or conflict regions.

In the disease arrival time analysis, all models that use the effective distance perform similarly well, including all gravity models with power-law distance decay. The disease arrival time $t_A$ correlates with the logarithm of the estimated import probabilities, i.e. the results should be in agreement with the logcorr goodness of fit results. The models with effective distance vary only by maximal 0.07 in their logcorr measures and these are based on 183 countries as potential source (Fig J in S1 Text). However, the 20 diseases in the arrival time analysis have only 10 unique outbreak countries. Additionally, due to factors like varying testing rates between countries, the uncertainty in arrival times, and other factors, the sample size is likely insufficient to recover the logcorr results. In order to decrease the noise on $t_A$, we repeated the analysis by extrapolating the arrival time via a logarithmic fit on the early cases, i.e. assuming an initial exponential growth (see Supplementary Note D in S1 Text). As a result of this procedure, some countries with insufficient data for extrapolation had to be excluded, which in turn led to the exclusion of more diseases. Nevertheless, the results are consistent with the $t_A$ estimation by 1st count (compare Fig 6 and Fig P in S1 Text).

We found that without providing any geodesic distance information to the import risk model, a distance dependence is recovered that is stronger for import probabilities to a single target, than from a single source, even if the import probability is computed from a source-centric view. Since the WAN is spatially embedded and has a network dimension of three [58], its connections reflect up to a certain degree the characteristics of the embedding space. This explains the import risk model's ability to capture distance dependence in general. That distance is a better predictor in the target-centric view aligns well with a previous study in which a target-specific human-mobility model collapses mobility data to multiple targets by assigning each target a specific attractiveness that is proportional to the target's population [51].

The import risk model predictions worsen for countries with a small outflow on the WAN, and since the country's WAN outflow is proportional to its gross domestic product, the model performs less good for countries with a lower GDP, i.e. small population and/or low to middle income countries. This is unfortunate, as our model derives Origin-Destination (OD) information (costly to directly monitor) from cost-effective traffic flow monitoring, making it particularly valuable for regions with limited resources. However, we find that the model alternatives (gravity, radiation) also perform poorly for low-outflow countries and that the passenger distribution of the latter is most likely constrained by the GDP and thus limited to the target-regions in effective proximity. To circumvent this problem, one could aggregate neighboring low-outflow countries until the conglomerate crosses the outflow threshold of $F_C = 10^6$ above which we observe a performance saturation (Fig 7 and Fig M in S1 Text). Of course, this compromise comes with a lower spatial resolution and we emphasize the need for future research in this direction.

While we have assessed the model's performance on the world air transportation network, its applicability extends to other modes of transportation such as subway systems, cars, buses, and trains. Future research will explore the specific conditions under which this model can be effectively applied. Furthermore, there is room for improvement in the basic estimation of the traveling population within an airport's catchment area based solely on its outflow. This

estimation does not currently account for the significant role of hubs and the missing information about transit passengers. The simple framework that only relies on the traffic network is appealing, but in certain scenarios its prediction can be refined by using information about the GDP, Gini-coefficient or population density.

Our comparison focused on the parameter-free radiation model and the fitted gravity model, but we acknowledge the existence of promising variations and alternative models that were not included in this study [30, 31, 54, 59]. However, the gravity model is widely applied and has been shown to perform equally well [59] or better than alternatives [54]. There are exceptions, e.g. an iterative computation of a gravity-like model outperforms the common gravity model in cases where the complete mobility network is not available [29]. Additionally, the radiation model outperforms the gravity model for long-distance connections [38, 54]. Still, the simplicity of the gravity model and its adaptability by parameter adjustment make it a strong counterpart. The model alternatives make use of the WAN-structure information by using the effective distance as done in e.g. Ren et al. [60] where the radiation model with time-distance was better than the travel-distance on the road network to predict the traffic on each link. Similarly, we observed that the effective distance, which is related to the arrival time of diseases, outperforms geodesic path-distance in predicting import probabilities.

The import risk model is fundamentally different from classic approaches that estimate OD trips from traffic data, because the latter find the OD trips that best reproduce the traffic data [28, 30, 44, 45], while our model runs a distribution process on the traffic data network. Thus, our model is a mechanistic bottom-up approach, while the classic approaches either fit and require the knowledge of the reference trip data [28, 30] or are based on the assumption that the trip distribution across the links follows the maximum entropy principle, i.e. the OD trips are considered as most likely that can be realized by the largest number of microstates [44, 45]. Note that maximum entropy approaches require an estimation of routes and their alternatives between each OD pair, while we allow all routes to be taken by the random walker. To the best of our knowledge, our model stands as unique in its mechanistic nature, enabling the study of modifications to its underlying distribution process. This includes strategies for containment aimed at slowing or restricting a pandemic, for instance. A straight forward implementation could be the testing of a fraction of passengers $C_i \leq 1$ at every transit airport $i$, which corresponds to reducing the probability to continue walking of an infected passenger (Eq 2) to

$$\tilde{S}_{n,n-1}(n_0, \mathbf{C}) = (1 - C_{n-1}) \times P_{n,n-1}(1 - q_{n-1}(n_0)) \ .$$

With $\mathbf{C} = [C_1, C_2, \ldots]$ one could allow for a varying testing capacity between the airports.

## Material and methods

### Data sources

The WAN provided by OAG (Official Airline Guide) [46] contains the number of flights and the respective maximum seat capacity $F_{i,j}$ between airports $i$ and $j$ aggregated for the year 2014. The reference import probability $\hat{p}(m|n) = \hat{T}_{mn}/\hat{T}_n$ is based on the "Global Transnational Mobility Dataset" [40, 47] that assigns the number of trips in 2014 $\hat{T}_{mn}$ from country $n$ to $m$ worldwide by combining the world air transportation origin-final-destination data set from the company SABRE, and cross-boarder visits with an overnight stay from the UNWTO (World Tourism Organization). Thus, $\hat{p}(m|n)$ represents not only the mobility via air travel but also via other means (sea, road, rail). However, air travel dominates long distance trips which makes it a fair reference set of the air transportation origin-final-destination matrix. For details on how the data sets were combined, see Supplementary Note A in S1 Text.

## Alternative models

The **gravity model** states that the number of trips between regions $n$ and $m$ increase with their population sizes ($N_n$ and $N_m$) and decrease with distance $d_{nm}$

$$T_{mn} = O_n \frac{N_n N_m}{f(d_{nm})} \ , \tag{10}$$

with $f(d)$ as a function that grows monotonically with distance $d$, most often chosen as either a power-law $f(d) = d^\gamma$ or an exponential $f(d_{nm}) = e^{\gamma d}$.

In the **radiation model**, the trips from $n$ to $m$ depend on their respective population sizes $N_n$, $N_m$ (or other measures as job opportunities) and on the number of people $s_{mn}$ that are in a circle with radius $r_{mn}$ centered around location $n$ including $N_n$ and $N_m$:

$$T_{mn} = O_n \frac{N_n N_m}{(s_{mn} - N_m)s_{mn}} \ . \tag{11}$$

The import probability of both models is computed by normalizing the trips with respect to the source-region

$$p(m|n) = \frac{T_{mn}}{\sum_j T_{jn}} = \frac{T_{mn}}{T_n} \ . \tag{12}$$

## Trip-symmetrization

We correct the import probability via symmetrizing the OD-matrix by (i) compute the estimated OD-matrix

$$T_{m,n}^{(0)} = p^{(0)}(m|n)N_n \tag{13}$$

from the import probability estimate, (ii) correct it by computing its symmetric part

$$\mathbf{S} = (\mathbf{T} + \mathbf{T}^\top)/2 \tag{14}$$

and (iii) compute the corresponding corrected import probability via

$$p^{(1)}(A|B) = \mathbf{S}_{AB}/S_B \ . \tag{15}$$

By going through these steps, the asymmetry is reduced heavily but still persists. Thus, we repeat steps (i) till (iii) until $p^{(3)}(A|B)$, which returns for all models a comparable asymmetry in mean and median to the reference data (see Supplementary Note B in S1 Text for details).

## Comparison measures

We compare the import probability models with the reference data via the Pearson correlation

$$\text{corr}(x, y) = \frac{\text{E}[(x - \bar{x})(y - \bar{y})]}{\sigma_x \sigma_y} \ , \tag{16}$$

with $\text{E}[x] \equiv \bar{x}$ as average, the root-mean-square error

$$\text{RMSE}(x, y) = \sqrt{\text{E}[(x - y)^2]} \ , \tag{17}$$

the common part of commuters [59]

$$\text{cpc}(x, y) = \frac{2\sum_{ij} \min(x_{ij}, y_{ij})}{\sum_{ij} x_{ij} + \sum_{ij} y_{ij}} \quad, \tag{18}$$

which is 1 if all links are identical and 0 if none of them agrees. All the above measures are more sensitive to strong links, i.e. large import probabilities. However, if the focus is to get a fair comparison including all links, we are more interested in logarithmic versions of the above measures or rank correlations. Thus, we compare the logarithm of the import probabilities via correlation

$$\text{logcorr}(x, y) = \text{corr}(\log(x), \log(y)) \quad, \tag{19}$$

root-mean-square error

$$\text{logRMSE}(x, y) = \text{RMSE}(\log(x), \log(y)) \quad, \tag{20}$$

and use the Kendall rank correlation coefficient

$$\tau_{\text{Kendall}} = \frac{C - D}{\sqrt{(C + D + T_x)(C + D + T_y)}} \quad, \tag{21}$$

with $C$ and $D$ as the number of concordant and discordant pairs and $T_x$ and $T_y$ as ties only in $x$ and $y$, respectively.

To simplify and generalize the comparison we combine the six above defined measures by computing the mean rank of each model, i.e. the best correlating model has the highest (12) and the worst the lowest (0) rank and the mean rank of one model is the average of all six ranks.

To quantify the mean difference between the models we define the relative performance of one model $M$ as

$$\text{rel.perf.}(f(x_M, y)) = \frac{f(x_M) - \text{worst}(f(x_k), k)}{\text{best}(f(x_k), k) - \text{worst}(f(x_k), k)} \quad, \tag{22}$$

with $f(x_M) = f(x_M, y)$ as the specific comparison function and $\text{best}(f(x_k), k)$ and $\text{worst}(f(x_k), k)$ as the best and worst performing value of all models using this comparison function. Note, that $\text{best}(\ldots) = \max(\ldots)$ apart for the rmse-measures, where it is $\min(\ldots)$ (analog for worst $(\ldots)$).

## Disease arrival times

The disease arrival time $t_A(i)$ in country $i$ is estimated by the date of the first reported case for H1N1 and SARS-CoV-2. For the SARS-CoV-2 variants we use the first sequenced sample in this country. However, for certain variants some sequenced samples appear in the statistics month before the outbreak date declared by the WHO [61], which we treat as misclassifications, discard them and use instead the first sample after the WHO listed outbreak for the respective country (see Supplementary Note D for details and Fig O in S1 Text). For each of the diseases/variants we used the WAN that we have access to and that is closest to the respective outbreak date (see Table B in S1 Text) and as outbreak country we used the one listed by the WHO as first country with first sequenced sample of the respective variant [61]. For the H1N1 outbreak in 2009 we used the case data provided by FluNet [62, 63] (the column *AH1N12009*), for the COVID-19 cases we use the WHO COVID-19 dashboard [64] accessed

through ourworldindata.org, the number of sequenced samples was accessed through GISAID [65–67] using the file *gisaid_variants_statistics.json*.

## Supporting information

**S1 Text.** Supplementary Note A: Origin-destination data ("Global Transnational Mobility Dataset"). Supplementary Note B: Symmetrized flows. Supplementary Note C: On the overestimation of low import probabilities. Supplementary Note D: Disease arrival time analysis. **Table A**. Filtering criteria for the log-cases fit to extrapolate the arrival times $t_A$. A country is excluded if (C0:) the detection is too sparse before peak-0 (less than 6 weeks of data), (C1:) the number of cases at peak-0 is below 30 (otherwise the signal is too noisy), (C2:) the extrapolated arrival time is before the WHO-outbreak date. $N$ is the number of countries for which case data could be generated. $N_{C0}$ and $N_{C1}$ are the countries that pass criteria C0 and C1. $N_{C0 \& C1}$ and $N_{C0 \& C1 \& C2}$ are the numbers of countries that pass multiple criteria. **Table B**. **Disease and SARS-CoV-2 Variant outbreak information and WAN date**. For each disease/variant the outbreak country and the date of the WAN used to compute the import probability estimates with the different models is displayed. Note that we only have the WAN from the years 2014 and 2019 in a yearly resolution and from 2020–2022 in monthly resolution. We repeated the analysis for COVID with the WAN from the month 2020–01-01, instead of using the yearly WAN from 2019, which gave comparable results. **Fig A**. **Import probability dependence on** the geographic distance (**A**), the effective distance (**B**) and the geographic path distance (**C**). The orange line represents the median and $C(x, y)$ is the correlation between the two measures either log-transformed or not. The geographic distance between countries is averaged over all airport pairs. The geographic path distance is the geographic distance along the shortest path derived from the WAN using $d_{\text{eff}}$, i.e. it is a combination of geographic and network information. The axis scale corresponds to the one with the highest correlation, i.e. log-log for distance and path distance (**A**, **C**) and y-log for the effective distance (**B**). **Fig B**. **Gravity model scans**. Parameter dependence of measures that compare the model estimated import probability with the reference import risk $\hat{p}(i|n_0)$. Thereby is "corr" the correlation, "cpc" the common part of commuters, "log_corr" the correlation on log-scale, "rmse" the root mean squared error and "kendalltau" the rank correlation via Kendalls tau. Two versions of the gravity model are shown with an exponentially decaying distance function $f(d) = e^{-\gamma d}$ (left column: **A, C, E**), and a power law decaying distance function $f(d) = d^{-\beta}$ (right column: **B, D, F**). As distance the geodesic distance (first row: **A, B**), the geodesic path distance (second row: **C, D**) and the effective distance (third row: **E, F**) are used. The dotted horizontal lines show the comparison measure with the import risk as model and have the same respective color. **Fig C**. **Mean optimal parameters for gravity models**. For each gravity model with exponentially and power law decaying distance function and with one of the three different distance measures (geodesic distance, geodesic path distance and effective distance), the exponent $\gamma$ or $\beta$ that results in the best fit to the reference import risk is shown. The comparison is quantified via the correlation (corr), correlation between the log-transformed import risks (log_corr), root mean square error (rmse), root mean square error of the log-transformed import risks (log_rmse), Kendall rank correlation (kendalltau) and the common part of commuters (cpc). The mean optimal parameter for each model is marked by a horizontal line and their values are $\gamma = [6.71, 6.41] *$ $10^{-4}$ for geographic and geo. path distance and $\gamma = 0.84$ for the effective distance, and $\beta = [1.90, 1, 95, 5.10]$ for geo., geo. path, and effective distance, respectively. **Fig D**. **Symmetry check for OD-matrix**. Each dot represents the number of passengers that travel between 2 countries and back. The OD-matrix is computed by the radiation model (1st. column), gravity model with exponentially (2nd column) and power law decaying (3rd column) distance

function and by the import risk model (4th column). The OD-matrix of the models is computed by multiplying the import probability with the source-outflow. The reference trips and return trips have the highest symmetry (5th column, **M**). The orange line depicts the median and the gray line is $y = x$ and illustrates perfect symmetry. The mean ($\mathrm{AVG}(a_{sym})$) and median ($\mathrm{MED}(a_{sym})$) asymmetry of the flows, computed according to Eq. C in S1 Text., are shown in each panel. The reference trips (**M**) show the lowest asymmetry, especially for large passenger flows. **Fig E**. **Relative comparison measures for the import probability estimates**. The rank (**A**) and the relative performance (**B**) for the different import probability estimation models. The model that agrees best (worst) with the reference import risk according a specific measure has the highest (lowest) rank and a relative performance of one (zero). The relative performance is then a linear interpolation between the best and worst model. The comparison measures are the correlation (corr), correlation between the log-transformed import risks (log_corr), Root-mean-square error (rmse), Root-mean-square error of the log-transformed import risks (log_rmse), Kendall rank correlation (kendalltau) and the common part of commuters (cpc). As exponents of the gravity models the mean optimal parameter is used (horizontal lines in Fig C in S1 Text.). **Fig F**. **Absolute comparison measures for the import probability estimates**. The comparison measures are the correlation (corr), correlation between the log-transformed import risks (log_corr), Root-mean-square error (rmse), Root-mean-square error of the log-transformed import risks (log_rmse), Kendall rank correlation (kendalltau) and the common part of commuters (cpc). As exponents of the gravity models the mean optimal parameter is used (horizontal lines in Fig C in S1 Text.). The colors depict the 4 different models. The solid circles are the models with corrected import probability by symmetrizing their OD-matrix, and the transparent squares are the non-corrected import probabilities of the respective model. **Fig G**. **Import risk comparison and its deviation from a linear relation**. Scatter plot (left) and only median and IQR with an exponential fit (right). **Fig H**. **Variations of the import risk model** to investigate how additional parameters influence the relation between the import risk and the reference import risk. **A**: the flow scaling exponent $v$ that estimates the travelling population $N(i)$ of the airport $i$ depending on its WAN outflow $F_i$ via $N(i) = F_i^v$ (default: $v = 1$). **B**: the effective distance offset $d_0$ that penalizes larger hop-distances in the effective distance $d_{\mathrm{eff}}(i|n_0) = d_0 - \ln(P_{ij})$ when creating the shortest path tree (default: $d_0 = 1$). **C**: the descendant fraction introduced in the shortest path exit probability, where 0.5 is the default value and values larger than 0.5 mean that the exiting at the descendant (or offspring) nodes compared to the current node becomes more likely. **D**: different weight options introduced for the shortest path tree exit probability. Per default, the node populations are not weighted. The weight is the inverse of either the geodesic or the effective distance. **E**: manually set shortest path exit probability of leaf nodes (dead-end nodes). Per default, the exit probability is 1. A decrease to 0.9 or 0.8 does not visually change the median. **Fig I**. **Country outflow reconstruction by import risk**. The flow in the WAN leaving a country $F_C$ is estimated by the import risk model by $T_C = \sum_{n \in C} \sum_{m \notin C} p_\infty(m|n) N_n$. Both measures are directly compared (**A**) and the relative error is computed depending on the number of airports in the respective country $N_{arpts}$ (**B**). The import risk model does not include the concept of a country which partly explains the overestimation for larger airports. Another explanation is the overestimation of the respective airport population $N_n = F_n$ by the WAN outflow for the import risk model (the true population is smaller because of the transit passengers that need to be excluded). Note that the WAN is used here, i.e. we check for self-consistency of the model and no reference data is included. **Fig J**. **Uncorrected models: rank and relative performance**. Same analysis as in the main text in Fig 4), however, here the uncorrected model predictions are used, i.e. without symmetrizing the OD-matrix. **Fig K**. **Mean correlation between arrival time and effective model distance vs. the speed of the disease** estimated by the mean arrival

time $\langle t_A(C)\rangle_C$, averaged over all countries $C$. The correlation $C(t_A, d_M)$ between arrival time $t_A$ and effective model distance $d_M$ is averaged over all models. The size of the datapoints illustrates the number of countries that were reached by the disease. **Fig L**. **Import risk between world regions to a specific target region**. In contrast to its derivation the import risk is displayed in a target-centric view, i.e. each panel displays the import probability to a single target region from all source regions. The distance between world regions is the mean distance between their airport locations. The grey line represents a power-law fit $p_\infty = c \cdot d^{-\alpha}$. The mean import risk is marked for each world region by a horizontal dashed line. The 22 target-world-regions are sorted according to their mean import risk. Maps are created with geopandas [48]. **Fig M**. **Source countries prediction quality and WAN outflow for two gravity models**. Same model-result representation as in Fig 7 but here instead of the import risk model, the gravity model with power-law distance decaying function using the geodesic $d_{\mathrm{geo}}$ (left) or effective $d_{\mathrm{eff}}$ (right) distance is applied. Also for these models the logcorr between import probability estimates $p(i|n_0)$ and the reference data $\hat{p}(i|n_0)$ improves for countries with a larger outflow in the WAN. **Fig N**. **WAN flow out of countries vs. population and GDP** The WAN flow out of a country is best mapped by its gross domestic product (GDP, **C**) compared to its population (**A**) or per capita GDP (**B**). The linear double-logarithmic regression results are shown in the lower part of each panel (r- and p-value). The size of each country corresponds to its population (**A**) and the color codes its continent. GDP is taken from the World Bank Dataset for the year 2014 [69]. **Fig O**. **Variant outbreak detection and fraction of sequenced samples** for each of the considered variants. To illustrate the spread of the variant and how often it occurs worlwide the fraction of the variant in all sequenced probes is plotted, i.e. if it reaches 1, all sequenced probes are the respective variant. The official WHO outbreak date [61] is highlighted as red dotted vertical line. We estimated an outbreak date by 45 days before the fraction of sequenced samples reached 2.5% of its world-wide peak. The orange vertical lines (lower row of lines) show for each country the arrival of the variant, estimated by the first sequenced probe ("count1"). The black vertical lines (upper row of lines) show the arrival times after the outbreak which are used in the main text. **Fig P**. **Correlation analysis with log-cases estimated arrival time**. Each model's import probability is converted to an effective distance $d_M(i|n_0) = -\ln(p(i|n_0))$ with $n_0$ as the outbreak country of the respective disease. The correlation results $C(t_A, d_M)$ with the arrival time $t_A(i)$ of the disease in the target country $i$ are grouped by model (**A**) and by the disease (**B**). As comparison distances, the correlation of the geodesic, geodesic path (on the effective shortest path tree) and the effective distance with $t_A$ are shown. Each dot represents a correlation result of the 10 considered outbreaks (H1N1 in 2009, COVID-19 in 2020 and the spread of 8 of its variants in the years 2020–2022). For the analysis only those diseases/variants were used with more than 10 datapoints (see Table A in S1 Text.). **Fig Q**. **New case numbers of the Alpha variant for countries that passed the selection criteria** for the log-cases fit to extrapolate the arrival time $t_A$ in the attempt to reduce noise. The vertical dashed line marks the outbreak as listed by the WHO [61], the yellow star is the extrapolated arrival time from the log-cases fit that is illustrated by a yellow line. To determine the peak-0 (marked by a vertical line) we used a difference analysis on the smoothed new-cases data. **Fig R**. **New case numbers of the Alpha variant for countries that failed the selection criteria** for the log-cases fit to extrapolate the arrival time $t_A$ in the attempt to reduce noise. The vertical dashed line marks the outbreak as listed by the WHO [61]. Those countries that passed the criteria C0 and C1 (see Table A in S1 Text. for details) show the log-cases fit. Note that the latter have an extrapolated $t_A$ before the outbreak date listed by the WHO. To determine the peak-0 (marked by a vertical line) we used a difference analysis on the smoothed new-cases data.
(PDF)

## Acknowledgments

We acknowledge Marc Wiedermann for insightful comments.

## Author Contributions

**Conceptualization:** Pascal P. Klamser, Dirk Brockmann.

**Data curation:** Pascal P. Klamser, Adrian Zachariae, Benjamin F. Maier, Olga Baranov, Clara Jongen, Frank Schlosser.

**Formal analysis:** Pascal P. Klamser, Adrian Zachariae, Benjamin F. Maier.

**Funding acquisition:** Dirk Brockmann.

**Methodology:** Pascal P. Klamser, Benjamin F. Maier, Frank Schlosser, Dirk Brockmann.

**Software:** Pascal P. Klamser, Adrian Zachariae, Benjamin F. Maier.

**Visualization:** Pascal P. Klamser.

**Writing – original draft:** Pascal P. Klamser.

**Writing – review & editing:** Pascal P. Klamser, Adrian Zachariae, Olga Baranov, Clara Jongen, Dirk Brockmann.

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
