## [Decision Letter · Decision Letter 0]

6 Jul 2023

Dear Mr. Klamser,

Thank you very much for submitting your manuscript "Inferring country-specific import risk of diseases from the world air transportation network" for consideration at PLOS Computational Biology. As with all papers reviewed by the journal, your manuscript was reviewed by members of the editorial board and by several independent reviewers. The reviewers appreciated the attention to an important topic. Based on the reviews, we are likely to accept this manuscript for publication, providing that you modify the manuscript according to the review recommendations.

Sincerely,

Yamir Moreno

Academic Editor

PLOS Computational Biology

Thomas Leitner

Section Editor

PLOS Computational Biology

Reviewer's Responses to Questions

**Comments to the Authors:**

Reviewer #1: Review attached

Reviewer #2: In this study, the authors describe a model to compute import probabilities in the early stage of an epidemic based on the topological structure of the worldwide airport network.

The paper tackles an important research problem with significant implications for public health. It is clear that accurate estimates of import probabilities at the beginning of a pandemic represent an invaluable asset for policy makers.

Overall, the paper represents a relevant study whose results will be of interest to the readers of PLOS Computational Biology.

Before recommending the paper for publication, I highlight three main concerns with the hope that these will help improve the manuscript.

1-Background

I think the manuscript lacks some relevant context regarding the importance of estimating arrival times and importation probabilities in pandemic management.

First, I am a bit puzzled by the sentence in the abstract: “Accurate mechanistic models to estimate such risks are still lacking”. Here, it seems that we completely lack models to estimate importation risks in accurate ways, but I think this is not completely true.

For instance, the GLEAM model (Balcan, D., et al. 2010. Journal of Computational Science, 1, 132–145.) has been extensively used to compute importation risks for different outbreaks/pandemics. Just to give an example, please have a look at the platform https://epirisk.net.

Also, in the reference: Piontti, Ana Pastore Y., et al. "The infection tree of global epidemics." Network Science 2.1 (2014): 132-137, the authors provide a description of how to estimate the disease importation risk the worldwide air travel network.

Finally, importation probabilities and arrival times from Mexico have been used at the beginning of the 2009 H1H1 pandemic to estimate R0 (Tizzoni et al. BMC Medicine 2012) or to evaluate the effects of travel restrictions on the international spread of the 2014 WA Ebola epidemic (Poletto et al. Eurosurveillance 2014).

I understand the model presented in this paper extends and improves previous approaches, as it computes import probabilities considering all possible routes of travel, but still I think the Introduction could be rephrased to better contextualize the study.

2-The radiation model.

I see little justification to consider the radiation model as a good candidate model to represent worldwide movements at the intercontinental scale. The model was not originally developed to represent movements across countries, and especially when we consider countries in different continents. The theoretical framework of intervening opportunities is hard to apply to the case of the international air travel, given the presence of oceans, that are not populated. Its poor performance for this specific case does not surprise me.

In short: has ever the radiation model been used to model the WAN? If not, why using it here? What is the reason, besides its popularity? This choice would require more justification and results should be contextualized given the clear advantage of the gravity model.

3- Real-world cases.

The theoretical analysis of the paper is well presented; however, I see one main limitation of the study in the fact that it does not consider any real-world epidemic to demonstrate the benefits/merits of the approach.

I think the paper would be significantly strengthened by an additional analysis considering a real epidemiological scenario, for instance the case of the COVID-19 pandemic.

What were the importation risks from Wuhan in December 2019? We know European countries were affected first, especially the case of Italy as one of the first countries to experience a major epidemic.

Does the model capture the pattern of case importations we observed in the early stage of the pandemic?

Does it provide insights that are new with respect to what could be done with simpler approaches?

**Have the authors made all data and (if applicable) computational code underlying the findings in their manuscript fully available?**

Reviewer #1: Yes

Reviewer #2: Yes

PLOS authors have the option to publish the peer review history of their article (what does this mean?). If published, this will include your full peer review and any attached files.

Reviewer #1: No

Reviewer #2: No

Figure Files:

Data Requirements:

Reproducibility:

References:

---

## [Decision Letter · Decision Letter 1]

21 Dec 2023

Dear Mr. Klamser,

We are pleased to inform you that your manuscript 'Inferring country-specific import risk of diseases from the world air transportation network' has been provisionally accepted for publication in PLOS Computational Biology.

Best regards,

Yamir Moreno

Academic Editor

PLOS Computational Biology

Thomas Leitner

Section Editor

PLOS Computational Biology

Reviewer's Responses to Questions

**Comments to the Authors:**

Reviewer #1: ALL MY PREVIOUS COMMENTS WERE DULY ADDRESSED.

Reviewer #2: I thank the authors for their extensive revision and the additional experiments they carried out. I believe the paper can be accepted for publication in its current form.

**Have the authors made all data and (if applicable) computational code underlying the findings in their manuscript fully available?**

Reviewer #1: Yes

Reviewer #2: Yes

PLOS authors have the option to publish the peer review history of their article (what does this mean?). If published, this will include your full peer review and any attached files.

Reviewer #1: **Yes: **Daniel Andrés Díaz-Pachón

Reviewer #2: No

---

## [Editor Report · Acceptance letter]

15 Jan 2024

PCOMPBIOL-D-23-00703R1 

Inferring country-specific import risk of diseases from the world air transportation network

Dear Dr Klamser,

I am pleased to inform you that your manuscript has been formally accepted for publication in PLOS Computational Biology. Your manuscript is now with our production department and you will be notified of the publication date in due course.

With kind regards,

Anita Estes
